# Targeting 5-HT Is a Potential Therapeutic Strategy for Neurodegenerative Diseases

**DOI:** 10.3390/ijms252413446

**Published:** 2024-12-15

**Authors:** Cencan Xing, Hongyu Chen, Wangyu Bi, Tong Lei, Zhongci Hang, Hongwu Du

**Affiliations:** 1Daxing Research Institute, University of Science and Technology Beijing, Beijing 100083, China; cencanxing@ustb.edu.cn (C.X.); 13907592802@163.com (H.C.); biwangyu@163.com (W.B.); zhongci-hang@xs.ustb.edu.cn (Z.H.); 2School of Chemistry and Biological Engineering, University of Science and Technology Beijing, Beijing 100083, China; tonglei1104@163.com

**Keywords:** 5-HT, clinical trials, neurodegenerative diseases, therapeutic target

## Abstract

There is increasing interest in the potential therapeutic role of 5-HT (serotonin) in the treatment of neurodegenerative diseases, which are characterized by the progressive degeneration and death of nerve cells. 5-HT is a vital neurotransmitter that plays a central role in regulating mood, cognition, and various physiological processes in the body. Disruptions in the 5-HT system have been linked to several neurological and psychiatric disorders, making it an attractive target for therapeutic intervention. Although the exact causes of neurodegenerative diseases such as Alzheimer’s disease (AD), Parkinson’s disease (PD), and amyotrophic lateral sclerosis (ALS) are not fully understood, researchers believe that regulating the 5-HT system could help alleviate symptoms and potentially slow the progression of these diseases. Here, we delve into the potential of harnessing 5-HT as a therapeutic target for the treatment of neurodegenerative diseases. It is important to note that the current clinical drugs targeting 5-HT are still limited in the treatment of these complex diseases. Therefore, further research and clinical trials are needed to evaluate the feasibility and effectiveness of its clinical application.

1 Introduction

Neurodegenerative disease (NDs) is a chronic, progressive disease that causes a reduction in the number of neurons in the nervous system and loss of function, severely impairing the patient’s cognitive and motor function. These diseases include Parkinson’s disease (PD), Alzheimer’s disease (AD), Huntington’s disease (HD), multiple sclerosis (MS), and amyotrophic lateral sclerosis (ALS), among others. In these diseases, patients often have varying degrees of anxiety, depression, and cognitive impairment [1,2,3]. The pathogenesis of NDs usually involves a variety of processes, such as abnormal protein aggregation, oxidative stress, and neuroinflammation [4,5,6]. At present, the clinical treatment of NDs mainly focuses on drug therapy, and most drugs cannot achieve a radical cure for NDs. Therefore, the scientific community is working hard to find new targets that can prevent NDs and develop more effective treatments. The serotonergic system is critical in key processes that regulate the central nervous system (CNS), such as in the brain, where the synthesis of neuroactive metabolites is regulated by serotonin-related metabolites [7], Tryptophan-related metabolic pathways are also implicated in CNS diseases [8].

5-hydroxytryptamine (5-HT, also known as serotonin), is a neurotransmitter synthesized from tryptophan in mammals and is widely distributed in most peripheral tissues and the brain [9]. As a key neurotransmitter in the CNS, 5-HT plays a crucial role in regulating emotions, sleep, and appetite through synaptic transmission between neurons, and it is closely associated with brain function [10]. An excess or deficiency of 5-HT may lead to various disorders, such as depression [11], anxiety [12], sleep disorders [13], and eating disorders [14]. Recent research discovered that 5-HT may also be related to NDs, such as PD, AD, and HD [8,15,16,17,18].

In summary, this review proposes the viewpoint that “5-HT can serve as a new target for the treatment of NDs”, (Figure 1) and this will be elaborated in detail below.

## 2. The System of 5-HT

### 2.1. Characteristics of the 5-HT System

5-HT receptors are generally classified into 7 families and 14 subtypes [19]. Among them, 5-HT_1_ receptors are subdivided into 5-HT_1A_, 5-HT_1B_, and 5-HT_1D_ subtypes, which are primarily distributed in the CNS and regulate the release of the 5-HT neurotransmitter. The 5-HT_1A_ receptor is associated with anxiety and depression [20], while the 5-HT_1B_ receptor is linked to appetite, sexual behavior, and temperature regulation [21,22,23]. The 5-HT_2_ receptors, which are divided into 5-HT_2A_, 5-HT_2B_, and 5-HT_2C_ subtypes, are mainly found in the cerebral cortex and hypothalamus. The 5-HT_2A_ receptor is related to hallucinations, delusions, and other psychotic symptoms [24]. The 5-HT_3_ receptor is primarily distributed in the CNS and the digestive system, and its agonists have emetic effects; thus, 5-HT_3_ receptor antagonists are used to treat nausea and vomiting [25]. The 5-HT_4_ receptor is mainly found in the brain and intestines, and its agonists can promote gastrointestinal motility and secretion, making them useful for treating intestinal motility disorders [26,27]. In addition to the aforementioned receptors, there are also 5-HT_5_, 5-HT_6_, and 5-HT_7_ receptors. The distribution and function of these receptors are not yet fully understood, but studies suggest they may be involved in memory, learning, sleep, and emotions. Below, we will introduce some common 5-HT receptors and write a table for easy reading (Table 1).

### 2.2. 5-HT Receptor and Its Function

#### 2.2.1. 5-HT_1_R

The 5-HT_1_R family includes 5-HT_1A_, 5-HT_1B_, 5-HT_1D_, 5-HT_1E_, and 5-HT_1F_ [28,29]. These receptors, part of the serotonin receptor family, are widely distributed in the central nervous system and mainly act as inhibitory receptors. They reduce neuronal excitability by lowering intracellular cAMP levels through G protein-coupled signaling [30]. As autoreceptors on presynaptic terminals, they inhibit further serotonin release when serotonin levels rise, creating a negative feedback mechanism to maintain neurotransmitter balance. Additionally, 5-HT_1_R also function as heteroreceptors, regulating the release of other neurotransmitters such as norepinephrine and dopamine, thus playing a key role in regulating neural pathways and maintaining neurotransmission balance [31]. Among these, 5-HT_1A_ is the most extensively studied subtype and is involved in various physiological and pathological processes. Its primary functions include the regulation of anxiety, mood, cognition, suicidal tendencies, appetite, sleep, and pain perception [32]. Currently, drugs targeting the 5-HT_1A_ receptor, such as buspirone, have been applied in clinical treatments, primarily enhancing the receptor’s function to exert anxiolytic effects. The 5-HT_1B/1D_ receptors are involved in regulating stress responses, mood, and motor control. Drugs such as naratriptan and sumatriptan are used to treat migraines by targeting these receptors [33,34,35]. The 5-HT_1E_ and 5-HT_1F_ receptors are involved in regulating physiological processes such as sleep, mood, and pain perception [29]. Lasmiditan, a 5-HT_1F_ agonist approved by the FDA in 2019 for the treatment of migraines, specifically suppresses the expression of calcitonin gene-related protein (CGRP). This selective mechanism reduces the likelihood of cerebrovascular side effects compared to other migraine treatments [36].

#### 2.2.2. 5-HT_2_R

The 5-HT_2_R family includes the 5-HT_2A_, 5-HT_2B_, and 5-HT_2C_ subtypes [28,29]. The 5-HT_2A_ receptor is one of the targets of hallucinogens and is involved in the regulation of cognition, mood, perception, and behavior [37]. Additionally, dopamine D2 and 5-HT_2A_ receptors have been shown to interact [38,39], playing a crucial role in neurotransmission. 5-HT_2A_ receptor antagonists, such as risperidone and olanzapine, are clinically used to treat schizophrenia, mood disorders, and other psychiatric conditions [40,41]. The 5-HT_2B_ receptor is expressed throughout the brain, particularly in the cerebellum, occipital cortex, and frontal cortex [42]. Recent studies found that 5-HT_2B_ receptor agonists may be useful therapeutic tools for improving depression treatment [43]. The 5-HT_2C_ receptor agonist lorcaserin is used to treat obesity [44].

#### 2.2.3. 5-HT_3_R

The 5-HT_3_R family has only one subtype, 5-HT_3_, which is the only known 5-HT ion channel receptor [28,29]. It is distributed in both the peripheral and CNSs and is involved in physiological processes such as the vomiting reflex, cognition, and anxiety [45]. Currently, clinical drugs targeting the 5-HT_3_ receptor subtype are primarily anti-nausea medications, such as ondansetron, granisetron, and palonosetron. These drugs mainly inhibit the function of the 5-HT_3_ receptor, thereby reducing nausea, vomiting, and other symptoms [46,47,48]. Among antipsychotic drugs, some also exhibit inhibitory effects on the 5-HT_3_ receptor, which may be one of the mechanisms through which they alleviate psychotic symptoms [49].

#### 2.2.4. 5-HT_4_R

5-HT_4_Rs are widely distributed in the CNS and peripheral organs. Their biological functions include regulating neurotransmission, influencing gastrointestinal motility and secretion, and participating in memory and learning processes [50,51]. Clinically, drugs targeting 5-HT_4_ receptors are primarily used to treat gastrointestinal diseases, such as alleviating dyspepsia, gastroesophageal reflux, and constipation. Currently listed 5-HT_4_ receptor agonists include cisapride, mosapride, and tegaserod [50,52,53]. Additionally, some studies found that 5-HT_4_ receptor agonists may have therapeutic effects on NDs and depression, but more research is needed to confirm these findings [54,55].

#### 2.2.5. 5-HT_5_R

The 5-HT_5_R family includes the 5-HT_5A_ and 5-HT_5B_ subtypes, primarily distributed in regions such as the hippocampus, hypothalamus, and amygdala. However, the functions and roles of these subtypes in clinical applications are not yet clear and require further research and exploration [56].

#### 2.2.6. 5-HT_6_R

The 5-HT_6_R family has only one subtype, 5-HT_6_, which is involved in regulating memory and learning, energy metabolism, and mood [57]. Currently, a drug targeting the 5-HT_6_ receptor, AVN-211, is in clinical trials. It acts as a 5-HT_6_ receptor antagonist and has great potential for treating AD [58].

#### 2.2.7. 5-HT_7_R

The 5-HT_7_R family includes the 5-HT_7A_ and 5-HT_7B_ subtypes, which are primarily distributed in CNS regions such as the hippocampus, hypothalamus, and amygdala. These receptors are involved in various biological functions, including learning and memory, mood regulation, sleep, addictive behaviors, and pain [59,60,61,62]. Activation of 5-HT_7_ receptors can enhance hippocampal neuron plasticity and learning and memory abilities, while their antagonists can affect these functions [63]. Current research on 5-HT_7_ receptors mainly focuses on antipsychotic drugs, anxiolytics, and sleep-regulating medications. Several 5-HT_7_ receptor antagonists, such as vortioxetine and lurasidone, improve mood, cognition, and sleep. Vortioxetine enhances serotonin transmission and cognitive function, aiding depression treatment [64]. Lurasidone, used for schizophrenia and bipolar disorder, improves mood and cognitive deficits by blocking 5-HT_7_ receptors, helping alleviate negative symptoms [65].

## 3. The Diagnostic and Therapeutic Potential of 5-HT in NDs

NDs such as PD and AD affect the function and survival of 5-HT neurons, leading to dysregulation of the 5-HT system [66,67]. Recent research highlighted the significant scientific value of 5-HT in the progression of these diseases: in the early stages or even before the onset of NDs, regulating the 5-HT system may help delay the occurrence and progression of the disease. Some 5-HT-related biomarkers can aid in diagnosing NDs; after diagnosis, monitoring indicators of 5-HT and its related metabolites can help assess disease progression and regulate its course. Additionally, modulating the 5-HT system can serve as an adjunct to existing treatments, improving therapeutic outcomes and patient quality of life.

Various factors can lead to dysfunctions in the 5-HT system, causing an imbalance in bodily functions. For instance, mutations in genes such as *SLC6A4* and *HTR1A* may lead to abnormalities in the 5-HT system, increasing the risk of depression, anxiety, and other mental disorders [68,69]. Stress, sleep deprivation, unhealthy diet, and lack of exercise can also affect the function of the 5-HT system [70,71,72]. Therefore, a deep understanding of the relationship between 5-HT and NDs can help us comprehend the pathology of these diseases and further explore therapeutic strategies for neurodegenerative disorders (Figure 2).

### 3.1. 5-HT Targeting AD Therapy

AD is caused by the extracellular deposition of beta-amyloid (Aβ) and the abnormal accumulation of hyperphosphorylated tau protein within brain cells, leading to neuronal death and loss of motor function [73]. In AD patients, there is a complex and regionally specific alteration in the expression and function of 5-HT receptors, these changes contribute to both the cognitive and neuropsychiatric symptoms of AD. Additionally, there is significant neuronal loss and dense neurofibrillary tangles in the raphe nuclei, which are responsible for producing 5-HT, in the brains of AD patients [74]. Activation of the serotonergic system can block the inflammatory response induced by Aβ oligomers in AD [75].

The main clinical treatments for AD include cholinesterase inhibitors and NMDA receptor antagonists [76,77]. Furthermore, new drugs such as antibodies against aggregated Aβ, BACE inhibitors that inhibit Aβ production, and tau aggregation inhibitors entered clinical trials [78,79,80]. Despite extensive research, currently available small-molecule drugs for AD have significant limitations: they only improve cognition temporarily and address only the symptoms of the disease without tackling its root causes [81]. These drugs primarily target two protein targets: inhibiting acetylcholinesterase (AChE), such as donepezil, or antagonizing NMDA receptors, such as memantine [82]. In 2021, the FDA approved the first disease-modifying option, the monoclonal antibody aducanumab, bringing hope for breakthroughs in AD treatment [83]. However, the FDA’s accelerated approval was mainly based on aducanumab’s ability to clear Aβ plaques, yet a correlation between Aβ clearance and cognitive impairment has not been demonstrated [84]. Additionally, treatment costs and safety concerns are prominent issues [85]. Therefore, there is an urgent need to discover novel small-molecule drugs targeting non-standard protein targets to meet this unmet clinical need.

Recent studies confirmed that 5-HT may influence the pathological features of AD by affecting oxidative stress and Aβ deposition. Compared to AD rats, those treated with 5-HT_1A_ receptor antagonists and 5-HT_2A_ receptor agonists showed significant biochemical improvements, including reduced levels of brain inflammation markers, oxidative stress indicators, and Aβ deposition [86]. Loratadine, a selective 5-HT_2A_ receptor antagonist, can alleviate inflammation by improving microglial dysfunction, promoting the clearance of neurotoxic substances, reducing Aβ deposition, and improving the pathological process in AD mice [87]. Additionally, 5-HT_4_ receptor agonists may prevent AD by enhancing acetylcholine release [88] and inhibiting Aβ secretion [89].

5-HT-related drugs may also improve the most prominent cognitive impairment issues in AD. 5-HT_6_ receptor antagonists can enhance cognitive function by modulating cholinergic neurotransmission: studies have shown that 5-HT_6_ receptor antagonists (such as idalopirdine) have the potential to improve memory and learning abilities in clinical trials. 5-HT4 receptor agonists can enhance acetylcholine release, which helps improve cognitive function and memory formation. Some AD patients may exhibit behavioral and psychological symptoms such as hallucinations and delusions; 5-HT_2A_ receptor antagonists, such as brexpiprazole and pimavanserin, have been shown to be effective in AD patients with severe psychosis. Brexpiprazole, in particular, acts through a more complex pharmacological profile, involving not only 5-HT_2A_ receptor antagonism, but also modulation of other neurotransmitter systems, which contributes to its efficacy in alleviating psychotic symptoms in these patients [90,91].

Additionally, 5-HT_6_ receptor antagonists can not only improve cognitive function in AD patients, but also potentially provide neuroprotection and anti-neuroinflammatory effects by regulating neurotransmitter release, improving synaptic function, and reducing neuroinflammation [92]. Novel multi-target ligands designed for the 5-HT_4_ receptor can bind to the 5-HT_4_ receptor, increase neuronal survival rates, reduce oxidative stress-induced cellular damage, and improve synaptic plasticity, offering new avenues for future AD treatment [93].

### 3.2. 5-HT Targeted PD Therapy

PD involves the loss of dopaminergic neurons, which may be caused by mitochondrial dysfunction and oxidative stress [94]. Before motor symptoms appear, PD can also cause emotional, cognitive, and psychiatric issues [95]. The hallmark of this disease is the early and significant death of dopaminergic neurons in the substantia nigra pars compacta (SNpc), leading to dopamine deficiency in the basal ganglia (BG) and progressing to the characteristic motor disorders of PD [96]. Additionally, the aggregation of alpha-synuclein is a key pathological feature of PD, contributing to the formation of Lewy bodies [97]. Increasing evidence suggests that, in addition to dopamine, 5-HT also plays a role in PD [98]. The 5-HT system originates in the raphe nuclei and projects to the BG, including the SNpc and caudate-putamen [9]. In PD patients, 5-HT neurotransmission is reduced in the late stages of the disease due to degeneration of the dorsal raphe nucleus [99]. Moreover, decreased levels of 5-HT, its metabolite 5-HIAA, and 5-HT transporters (SERT) have been found in the BG of some patients [100,101]. Notably, PD patients often exhibit emotional symptoms such as anxiety and depression before motor symptoms arise [102]. PET studies have shown that 5-HT neurotransmission dysfunction may be related to resting tremor [103]. As the disease progresses, the dopamine deficiency in the striatum and the 5-HT defect in the raphe nuclei may jointly contribute to the occurrence of resting tremor. Therefore, 5-HT treatment may be beneficial for PD patients with tremor related to raphe nucleus dysfunction [103].

To date, dopamine replacement therapy with levodopa combined with peripheral decarboxylase inhibitors remains the most effective treatment for alleviating motor symptoms of PD [104]. However, many PD patients experience motor fluctuations due to the diminishing effectiveness of the treatment over time [105], with the most severe being levodopa-induced dyskinesia [104,106]. Motor fluctuations can be improved by reducing the dose of levodopa, but this affects the control of PD motor symptoms. Therefore, there is an urgent need to develop new effective therapies to prevent and manage motor complications and dyskinesia associated with dopamine replacement therapy. Currently, treatment for all PD patients is symptomatic, focusing on improving both motor (e.g., tremor, rigidity, and bradykinesia) and non-motor (e.g., constipation, cognition, mood, and sleep) symptoms, with no available treatments that modify the course of the disease [107].

The vast majority of scholars believe that neuroinflammation and oxidative stress are closely related to the progression of PD. 5-HT_1A_ receptor agonists can alleviate levodopa-induced dyskinesia [108] and protect cells from MPTP-induced dopaminergic neurotoxicity [109], thus having neuroprotective potential. The 5-HT_1A_ and 5-HT_1B_ receptors also play roles in glial cell formation associated with PD and motor disorders: levodopa combined with 5-HT_1A/1B_ receptor agonists treated PD-induced motor disorders in rats and delayed the occurrence of neuroinflammatory markers in the nigrostriatal system [110]. Additionally, activating the 5-HT_1F_ receptor can promote mitochondrial biogenesis, reduce oxidative stress, and thus protect neuronal function and survival in PD mice [111]. Stimulating the 5-HT_1A_ receptor can promote astrocyte proliferation and upregulate antioxidant molecules [112], and 5-HT_6_ receptor antagonists can also protect astrocytes from damage [113]. The selective 5-HT_1A_ receptor agonist 8-OH-DPAT can protect hippocampal neurons from transient ischemia-induced damage in gerbils by regulating NMDA receptor NR1 subunit phosphorylation and increasing the expression of brain-derived neurotrophic factor (BDNF) [114].

Motor disorders significantly impact the quality of life for patients with PD. As PD progresses, common motor symptoms often reverse into L-DOPA-induced dyskinesia (LID), which may limit PD treatment [115]. In the 5-HT system, 5-hydroxytryptophan (5-HTP), a direct precursor of 5-HT, demonstrated significant anti-dyskinetic effects in PD rat models [116]. In PD mouse models, 5-HTP has been shown to reduce LID by modulating the AKT/mTOR/S6K signaling pathway [117], suggesting that 5-HT might have therapeutic effects on L-DOPA-induced motor disorders. Additionally, 5-HT_2A_ receptor antagonists have been shown to alleviate motor disorders in MPTP-induced PD mice [118]. Ondansetron, a 5-HT_3_ receptor antagonist, can alleviate L-DOPA-induced motor disorders without affecting the therapeutic efficacy of L-DOPA, indicating that selective 5-HT_3_ blockade may be an effective strategy for mitigating L-DOPA-induced motor disorders and slowing their progression [119]. Dopamine agonists used to treat PD can relieve motor symptoms but often cause side effects such as nausea and vomiting [120]. Since 5-HT receptors play a role in the emetic reflex, 5-HT-targeting drugs can help mitigate these side effects. As such, 5-HT drugs may be used in combination with dopamine agonists to alleviate the adverse effects of dopamine therapy, improving the overall treatment approach for PD [121].

In clinical studies, non-motor symptoms of PD also received considerable attention. Depression and anxiety are common emotional disorders in PD patients. Selective serotonin reuptake inhibitors (SSRIs) can help alleviate depressive symptoms in PD patients [122]. Additionally, some studies suggest that 5-HT_3_ receptor antagonists may be a potential treatment for depression and anxiety [123,124,125]. Research has shown that the 5-HT_2A_ receptor is associated with various psychiatric disorders, including psychosis. The highly selective 5-HT_2A_ receptor antagonist EMD-281,014 effectively reduces LID and psychosis-like behaviors in MPTP-lesioned common marmosets without affecting L-DOPA’s therapeutic effects on PD, demonstrating its potential as an anti-dyskinetic and antipsychotic strategy [126]. Selective 5-HT_2A_ antagonists/inverse agonists, such as Pimavanserin and M100907, can reverse psychosis-like behavior deficits in animal models of PD psychosis (PDP) without affecting motor behavior, suggesting that 5-HT_2A_ antagonism/inverse agonism may aid in the treatment of PDP [127]. Notably, Parkinson’s patients treated with the 5-HT_2A_ receptor antagonist Pimavanserin showed significant improvements in visual hallucinations and delusions [128]. Furthermore, pain is considered a significant non-motor manifestation of PD. Neuropathic pain is associated with changes in 5-HT levels [129]. Research indicates that inhibiting spinal 5-HT_3_ receptors and the excitability of spinal dorsal horn (SDH) neurons can reduce pain perception in PD rats [130]. Constipation and other gastrointestinal motility disorders also occur in many PD patients. Recent research found that electroacupuncture can significantly improve gut motility in PD mice, possibly through the stimulation of 5-HT_4_ receptor expression, which promotes the cAMP/PKA signaling pathway [131].

The functional involvement of 5-HT receptors in learning and memory may improve cognitive impairments in PD patients. For example, the 5-HT_7_ receptor is involved in memory deficits induced by long-term isoflurane anesthesia [132] and the 5-HT_4_ receptor also shows potential in learning and memory [133]. A recent study suggests that memory processing in rats is associated with the 5-HT system, which may involve 5-HT_1D_ and 5-HT_1F_ receptors in the hippocampus [134].

### 3.3. 5-HT Targeting ALS Therapy

ALS, also known as Lou Gehrig’s disease, is a progressive neurodegenerative disorder characterized by muscle weakness, atrophy, and stiffness, eventually leading to respiratory muscle weakness and death [135]. The etiology and pathogenesis of ALS remain unclear, but genetic variations and environmental factors are involved. Approximately 10% of cases are familial ALS (FALS), associated with genetic factors, while 90% are sporadic ALS (SALS) [135]. ALS primarily involves neuronal damage and death, and currently, there is no effective treatment for this devastating disease. Previous studies have shown that the pathogenesis of ALS is closely related to 5-HT [136,137,138]. While there is no cure for ALS, finding new therapeutic approaches is crucial. Although there is currently no evidence that 5-HT-targeted therapy can directly cure ALS, it has the potential to alleviate symptoms, improve quality of life, and reduce patient discomfort, which is significant for ALS patients.

The pathology of ALS mainly involves neuronal damage, glial cell abnormalities, neuroinflammatory responses, and abnormal protein aggregation. In these processes, 5-HT may act as a therapeutic target. Previous studies found that glutamate-induced neurotoxicity is triggered in rat nucleus accumbens neurons by reducing the binding of 5-HT to presynaptic 5-HT_1B_ receptors, and ultimately leading to the rapid loss of motor neurons [139], 5-HT can achieve neuroprotective and anti-inflammatory effects by modulating glutamate excitability [138,140]. The 5-HT_2B_ receptor is upregulated during ALS mouse spinal cord disease, and the deletion of the *HTR2B* gene encoding 5-HT_2B_R exacerbates the disease outcome in ALS mice [141]. Although treatment with a 5-HT_2B_ receptor agonist did not improve the ultimate disease outcome in ALS mice, it was found to modulate microglial gene expression, slowing the progression of ALS, suggesting that targeting 5-HT_2B_ might be a potential therapeutic approach [142]. After administration of 5-HT receptor antagonists, increased expression of TDP-43 (transcriptional regulator) and superoxide dismutase 1 (SOD1) proteins was observed, along with a significant increase in the number of astrocytes and microglia in certain anatomical regions and a reduction in the number of neurons. This may indicate that inhibition of certain 5-HT receptors promoted the onset of ALS [143]. Additionally, inflammation, including autoimmune inflammation, has been documented in the course of ALS, but treatments targeting inflammation have not yet been successful [144]. Previous studies have shown that the expression of the 5-HT_7_ receptor parallels the increased expression of TNF-α, IL-1β, and NF-κB, and 5-HT_7_ receptor agonists can produce anti-inflammatory effects [145], suggesting that 5-HT may slow the progression of ALS through anti-inflammatory mechanisms.

In the course of ALS, issues such as muscle spasms and pain, which significantly impact patients’ quality of life, need urgent solutions. Studies found that 5-HTP, a precursor of 5-HT metabolism, can increase 5-HT levels in ALS mice, improving motor function and survival rates [146]. The degeneration of 5-HT neurons may underlie spasms in ALS mice, and it has been found that 5-HT_2B/C_ receptor inverse agonists can eliminate spasticity symptoms, suggesting that these agonists may improve motor function in ALS [147,148]. Pain, including neuropathic pain and spasticity-related pain, is a common yet overlooked symptom in ALS patients. Existing research confirmed that several 5-HT receptors play crucial roles in the progression and regulation of central and peripheral pain, particularly 5-HT_1_, 5-HT_2_, 5-HT_3_, and 5-HT_7_ receptors [149]. Studying these receptors could potentially alleviate common pain symptoms in ALS patients. Sleep disruption is also prevalent among ALS patients [150] and 5-HT helps increase sleep propensity and regulate sleep disorders [151]. Additionally, early supplementation with 5-HT may alleviate or delay dysphagia symptoms, which occur in some ALS patients, thereby improving their quality of life [152].

Significant research has been conducted on survival prediction and disease prevention in ALS patients. Studies found that platelet 5-HT levels are associated with the survival of ALS patients, with higher platelet 5-HT levels correlating with longer survival, suggesting that platelet 5-HT levels may be a potential biomarker for predicting patient survival and are of great importance for the management and research of ALS [153]. Notably, research has shown that systemic injection of 5-HTP can delay hindlimb muscle weakness in ALS mice and reduce muscle tone. Additionally, high-dose administration of 5-HTP can significantly extend the lifespan of ALS model mice [138].

### 3.4. 5-HT Targeted HD Therapy

HD is an inherited neurodegenerative disorder affecting the nervous system, leading to the gradual loss of mental and motor functions, ultimately resulting in death [154]. It is caused by a defect in the Huntington gene (HTT) on chromosome 4, leading to the production of the Huntingtin protein. While this protein typically supports the development and function of the nervous system, in individuals with HD, it forms large abnormal aggregates within brain cells, eventually causing neuronal death and brain tissue degeneration [155]. Currently, there is no cure for HD, and only medications such as antipsychotics, antidepressants, anxiolytics, and psychotherapy can improve the quality of life for patients. Research has shown that there is a significant increase in SERT in the striatal tissue of grade 4 HD patients [156], and autopsy reports of HD patients also found significantly elevated levels of 5-HT [157], indicating a relationship between 5-HT and the pathogenesis of HD [156,158]. Thus, 5-HT is a potential therapeutic target for HD.

It is widely accepted that 5-HT receptor drugs can alleviate neuroinflammation, oxidative stress, and reduce apoptosis [159]. By modulating the 5-HT system, they may provide neuroprotective effects for HD patients. For example, N-acetyl-5-HT (NAS), a metabolite formed by the acetylation of 5-HT, has been shown to exert antioxidant and anti-apoptotic effects through the activation of the TrkB/CREB/BDNF pathway and the expression of antioxidant enzymes, demonstrating neuroprotective properties against oxidative stress-induced neurotoxicity [160]. HD patients often face severe motor disturbances, cognitive deficits, and psychological symptoms. Targeting the 5-HT system to regulate neurotransmission may improve these symptoms and enhance the quality of life for HD patients. Previous studies found that a single HD patient treated with perospirone, which acts as an antagonist for the 5-HT_2A_ receptor and an agonist for the 5-HT_1A_ receptor, showed significant control over involuntary movements and psychiatric symptoms [161], providing new avenues for future HD treatment research. Depression, anxiety, and irritability are common neuropsychiatric features in the preclinical stages of HD. Research indicates that altered 5-HT signaling underlies the occurrence of preclinical depression in HD [162], and several 5-HT receptors mediate anxiety behaviors [163]. Studies also found that many 5-HT receptors are associated with traits such as irritability and impulsivity [164].

### 3.5. 5-HT Targeted MS Therapy

MS is an autoimmune disease affecting the CNS, characterized by neuroprotective myelin damage and neuronal death. Myelin damage disrupts neural signal conduction, leading to motor, sensory, and cognitive issues. The etiology of MS is unclear, but likely involves genetic, environmental, and immune system factors [165]. Genes in the human leukocyte antigen (HLA) region may increase MS risk [166]. Sunlight, vitamin D, and geographic location also influence MS risk, with higher latitudes associated with greater risk [167]. Abnormal immune responses and infections may further contribute to MS development [168].

MS is a complex disease with limited treatment options. Previous studies noted reduced serum and platelet 5-HT levels in MS patients [169]. Among 5-HT receptors, the 5-HT_7_ receptor can activate initial T cells, influencing inflammatory responses. Research indicates that 5-HT_7_ is dysregulated in MS patients and can promote the release of IL-10, a potent immunosuppressive cytokine, suggesting that the 5-HT_7_ receptor may have immunoprotective effects [170]. Therefore, 5-HT_7_ can be considered an interesting therapeutic target for MS. Fluoxetine, a selective serotonin reuptake inhibitor, has been reported to have anti-inflammatory effects in MS [171]. Both fluoxetine and 5-HT can inhibit the production of IL-17, IFN-γ, and GM-CSF in stimulated CD4 T cells. Blocking the 5-HT_2B_ receptor alters these effects, while activating the 5-HT_2B_ receptor does not. These findings suggest that fluoxetine has anti-inflammatory effects in MS, possibly mediated through 5-HT_2B_ receptor activation [172].

Similar to ALS patients, MS patients also experience widespread neuropathic pain and spasms [173], significantly affecting their quality of life. Research found that 5-HT reuptake inhibitors have some effect on central neuropathic pain [174] and 5-HT_2B_ and 5-HT_2C_ receptors on motor neurons can lead to the recovery of motor neuron function and spasticity post-injury [175]. These findings suggest that 5-HT can alleviate pain in MS patients.

Furthermore, studies have shown that the incidence of depression and anxiety is higher in MS patients than in the general population [176,177]. In individuals with one or two short allele copies of the 5-HT transporter-linked polymorphic region (5-HTTLPR), a stronger association was found between the burden of major life events and the severity of MS-related depression. This implies that polymorphisms in the 5-HT transporter gene modify the relationship between major life events and depression in MS patients [178].

Currently, research on 5-HT as a therapeutic target for MS is in its early stages, and its efficacy and safety need further investigation and clinical trials. Although there is still insufficient evidence to support 5-HT as an effective treatment strategy for MS, studying 5-HT and its receptor drugs may reveal potential benefits for MS symptoms, providing direction and rationale for future clinical research and drug development.

### 3.6. The Potential of 5-HT as a Biomarker in Diagnosing NDs.

In the treatment of any disease, early diagnosis and prognosis are crucial, and searching for biomarkers to diagnose AD and assess neurodegeneration is a popular topic. Studies found that 5-HT levels in the brains of AD patients are reduced [179], and measuring levels of 5-HT and its metabolite, 5-HIAA, in cerebrospinal fluid may aid in diagnosing and monitoring AD progression [180]. PET imaging of 5-HT_1A_ receptors can evaluate mild neurodegenerative changes in AD [66] and lower 5-HT levels may be associated with poorer prognoses, such as cognitive decline and worsening emotional symptoms in both AD and PD patients.

For PD, effective clinical biomarkers are currently lacking. Research indicates that plasma levels of 5-HT and 5-HIAA are decreased in PD patients, suggesting these could serve as peripheral markers for non-motor symptoms of PD [181]. Additionally, reduced expression of the SERT is linked to the neurodegenerative process in PD. PET imaging has shown that the binding levels of SERT in the brains of clinically advanced, non-depressed PD patients are significantly reduced in multiple brain regions, indicating that serotonergic neuron damage may be a common feature in advanced PD [182].

Regarding HD, genetic testing is typically required for diagnosis, but early diagnosis and reliable biomarkers are essential for disease management and the development of treatment strategies. Recent studies have shown significantly reduced 5-HT levels in HD patients, indicating that this reduction could serve as a potential biomarker for HD, providing new directions for the diagnosis and treatment of HD [183].

## 4. Progress of Clinical Research on Targeting 5-HT for NDs

In the treatment of PD, the clinical application of 5-HT as a drug target is relatively limited. Pimavanserin, a popular drug, played a significant role in several clinical studies. Pimavanserin is a selective 5-HT_2A_ receptor inverse agonist primarily used to treat hallucinations and delusions in PD patients, particularly those associated with PDP. It alleviates these non-motor symptoms by reducing the activity of 5-HT_2A_ receptors. In addition to Pimavanserin, other studies explored the potential applications of various 5-HT system modulators in PD treatment.

Similarly, the clinical application of 5-HT-targeted drugs in AD treatment is also relatively limited. Currently, no specific 5-HT drugs have been approved for treating core symptoms of AD, such as cognitive decline and memory impairment. However, some studies are investigating methods to improve AD symptoms and disease progression by modulating the 5-HT system. These drugs mainly aim to alleviate AD-related behavioral and psychological symptoms, such as anxiety and depression, by modulating 5-HT receptors and influencing neurotransmitter activity. In the context of HD, ALS, and MS, 5-HT is rarely used as a drug target in clinical research. Current research primarily focuses on the potential of these drugs to improve motor behavior in these diseases.

The increasing number of clinical studies targeting 5-HT indicates that many institutions and research units can no longer overlook the significant role of 5-HT in the nervous system (Table 2). This recognition marks a new starting point for the treatment of NDs.

## 5. Conclusions

In conclusion, substantial research has been conducted on the role of 5-HT in NDs, but ongoing exploration is essential for further advancements. 5-HT exerts neuroprotective effects by promoting neuronal survival and function. It also has anti-inflammatory and immunomodulatory properties, regulating immune cell function and the extent of inflammatory responses. Furthermore, 5-HT plays a crucial role in modulating neural transmission through its interaction with receptors, which is significant in neurodegenerative conditions. Additionally, 5-HT influences neuronal differentiation, migration, and synapse formation, playing a key regulatory role in neural network formation and function.

Current studies indicate that 5-HT receptor agonists and antagonists may positively impact symptom relief and disease progression in NDs, potentially alleviating some side effects associated with current clinical treatments. This suggests that exploring the relationship between 5-HT and NDs can unveil new therapeutic strategies and drug targets. However, translating these findings into clinical outcomes remains limited.

While targeting the 5-HT system for treating NDs shows promise, it is important to note that it cannot yet serve as a standalone treatment method. The role of 5-HT in the brain is highly complex, and solely targeting 5-HT may lead to unforeseen side effects. Moreover, long-term use of 5-HT drugs can lead to tolerance, reducing their efficacy over time. Many potential 5-HT-targeting drugs struggle to cross the blood–brain barrier, limiting their effects on the CNS. Addressing these issues, future research could develop drugs that act on multiple neurotransmitter systems, reducing side effects associated with single-target treatments. For instance, modulators combining dopamine and 5-HT could offer more comprehensive therapeutic effects. Advanced drug delivery systems such as lipid nanoparticles, biomimetic nanoparticles, or other innovative methods could enhance the efficiency of crossing the blood–brain barrier.

Interestingly, studying the changes in 5-HT during neurodegenerative disease progression revealed potential biomarkers and indicators for early diagnosis and assessment. These biomarkers can significantly impact the treatment of NDs by monitoring disease progression, evaluating therapeutic efficacy, aiding clinical decisions, and potentially preventing disease onset. Consequently, more fundamental research is required to support these concepts, facilitating the translation of findings into clinical practice and genuinely improving patient outcomes and quality of life.

## Figures and Tables

**Figure 1 ijms-25-13446-f001:**
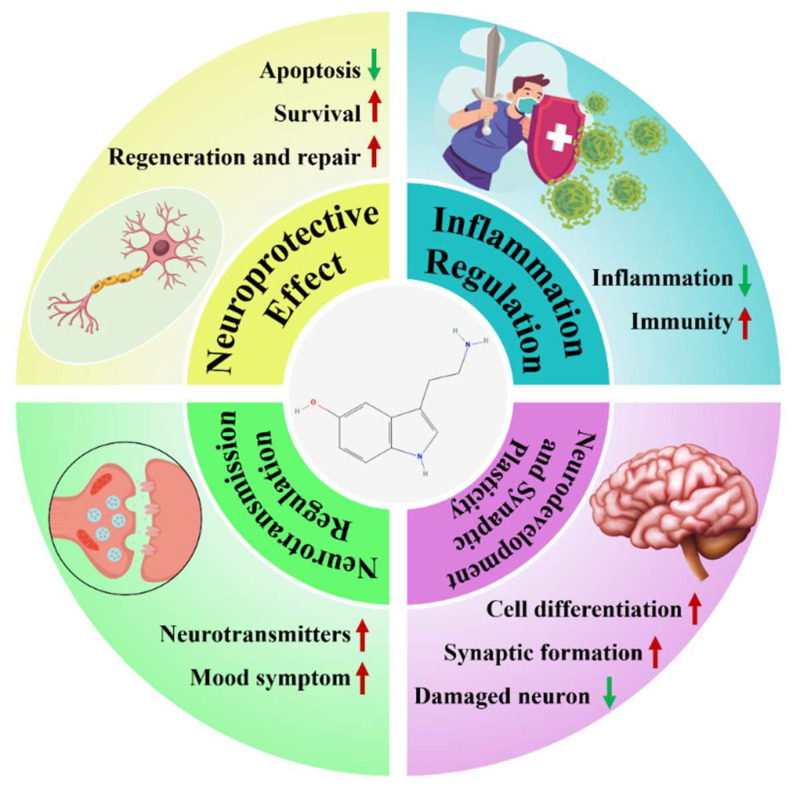
Summary of the therapeutic mechanisms of 5-HT in NDs. Red arrows, upregulation; green arrows, downregulation.

**Figure 2 ijms-25-13446-f002:**
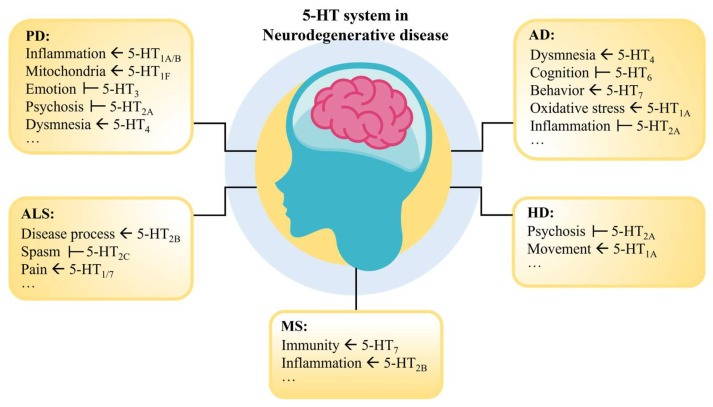
Function of the 5-HT receptor in NDs. Where “
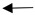
” indicates that the activation of this receptor contributes to the recovery of the symptom; “
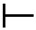
” indicates that inhibition of this receptor contributes to the recovery of the symptom.

**Table 1 ijms-25-13446-t001:** 5-HT receptor, function, and known agonists/antagonist.

5-HT Receptors	Possible Functions	Known Agonist/Antagonist
5-HT_1A_	anxiety, mood, cognition, suicidal tendencies, appetite, sleep, and pain perception	Buspirone
5-HT_1B/1D_	stress responses, mood, and motor control	Naratriptan
		Sumatriptan
5-HT_1F_	pain	Lasmiditan
5-HT_2A_	cognition, mood, perception, and behavior	Risperidone
		Olanzapine
5-HT_2C_	appetite regulation, weight control, mood, and behavior regulation	Lorcaserin
5-HT_3_	vomiting reflex, cognition, and anxiety	Ondansetron
		Granisetron
		Palonosetron
5-HT_4_	neurotransmission, gastrointestinal motility and secretion, and memory	Cisapride
		Mosapride
		Tegaserod
5-HT_6_	memory, energy metabolism, and mood	AVN-211
5-HT_7_	learning and memory, mood regulation, sleep, addictive behaviors, and pain	SB-269970
		SB-656104-A

**Table 2 ijms-25-13446-t002:** Clinical research of 5-HT in ND in the past 5 years. In status, “+” represents completed; “=” represents recruiting; and “−” represents not yet recruiting.

NCT Number	Stage	ND Type	Drug	Sample Size	Object	Mechanism	Year	Locations	Status
NCT03652870	III	Parkinson’s disease (PD)	Nortriptyline	52	Depression in PD	tricyclic antidepressants	29-Aug-18	Royal Free London NHS Foundation Trust, London, United Kingdom	+
			Escitalopram			selective serotonin reuptake inhibitor			
NCT03947216	II	Parkinson’s disease	Pimavanserin	130	Impulse control disorder (ICD) in PD	selective serotonin 5-HT_2A_ inverse agonist	13-May-19	Service de Neurologie-CHU Besançon, Besançon, France	=
NCT04292223	IV	Parkinson’s disease	Pimavanserin	29	Parkinson’s disease psychosis	selective serotonin 5-HT_2A_ inverse agonist	3-Mar-20	Movement Disorders Center of Arizona, Scottsdale, Arizona, United States	+
NCT04373317	IV	Parkinson’s disease	Pimavanserin	358	Parkinson’s disease psychosis	selective serotonin 5-HT_2A_ inverse agonist	4-May-20	Southern Arizona VA Health Care System, Tucson, AZ, Tucson, Arizona, United States	=
			Quetiapine			5-HT_2A_ receptor antagonist			
NCT05357612	IV	Parkinson’s disease	Pimavanserin	75	measure baseline 5HT_2A_ receptor density	selective serotonin 5-HT_2A_ inverse agonist	3-May-22	Vanderbilt University Medical Center, Nashville, Tennessee, United States	=
NCT05590637	IV	Parkinson’s disease	Pimavanserin	94	Parkinson’s disease psychosis, dementia with Lewy bodies	selective serotonin 5-HT_2A_ inverse agonist	21-Oct-22	UT Health Science Center—San Antonio, San Antonio, Texas, United States	=
			Quetiapine			5-HT_2A_ receptor antagonist			
NCT05796167	Early I	Parkinson’s disease	Pimavanserin	10	sleep quality in patients with PD and visual hallucinations/delusions	selective serotonin 5-HT_2A_ inverse agonist	3-Apr-23	−	−
NCT04167813	II	Parkinson’s disease	Ondansetron	306	Parkinson’s hallucinations, dementia with Lewy bodies	5-HT_3_ receptor antagonist	19-Nov-19	Grampian Aberdeen, United Kingdom	=
NCT04497168	II	Parkinson’s disease	Citalopram	58	reduce visuospatial cortex Aβ plaque accrual	Selective serotonin reuptake inhibitors	4-Aug-20	University of Michigan, Ann Arbor, Michigan, United States	=
NCT04932434	II	Parkinson’s disease	Psilocybin	10	depression and anxiety in PD	5-HT_2A_ receptor agonist	21-Jun-21	University of California, San Francisco, United States	=
NCT05148884	II	Parkinson’s disease	NLX-112	27	L-DOPA-induced dyskinesia in PD	high-efficacy selective 5-HT_1A_ receptor agonist	8-Dec-21	Sahlgrenska Hospital, Gothenburg, Sweden	+
NCT03724942	III	Alzheimer’s disease (AD)	Brexpiprazole	164	Agitation associated with dementia of the Alzheimer’s type	5-HT_1A_, 5-HT_2A_ receptor antagonists	30-Oct-18	Jisenkai Nanko Psychiatric Institute Shirakawa, Japan	+
NCT04123314	I	Alzheimer’s disease	Psilocybin	20	depression in mild cognitive impairment and early AD	5-HT_2A_ receptor agonist	10-Oct-19	Behavioral Pharmacology Research Unit, Baltimore, Maryland, United States	=
NCT04341467	−	Alzheimer’s disease	Olanzapine	76	Behavioral and psychological symptoms of dementia in AD	5-HT_2A_ receptor antagonist	10-Apr-20	Tianjin Anding Hospital, Tianjin, Tianjin, China	=
NCT05282550	II	Alzheimer’s disease	Trazodone	100	Amnestic mild cognitive impairment	5-HT_2A_ receptor antagonist	16-Mar-22	Johns Hopkins Hospital, Baltimore, Maryland, United States	=
NCT05397639	III	Alzheimer’s disease	Masupirdine	375	Agitation, Alzheimer’s type dementia	5-HT_6_ receptor antagonist	31-May-22	ATP Clinical Research, Inc. Costa Mesa, California, United States	=
NCT04071639	I	Huntington’s disease (HD)	Risperidone	60	Huntington’s disease motor, MMSE, psychiatric, and functional domains	5-HT_2A_ receptor antagonist	28-Aug-19	Second Affiliated Hospital, Zhejiang University School of Medicine, Hangzhou, Zhejiang, China	=
			Haloperidol			5-HT_2A_, 5-HT_2C_ receptor antagonists			
			Zoloft			Selective serotonin reuptake inhibitors			
NCT04302870	II/III	Amyotrophic lateral sclerosis (ALS)	Trazodone	800	Motor neuron disease	5-HT_2A_ receptor antagonist	10-Mar-20	Southern Health and Social Care Trust, Craigavon Area Hospital, Portadown, County Armagh, United Kingdom	=
NCT04546698	−	Multiple sclerosis (MS)	−	78	5-HT_7_ receptor implication in inflammatory mechanisms in MS	5-HT_7_ receptor	14-Sep-20	CHR Orléans, Orléans, France	+

## Data Availability

Not applicable.

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
