# Peer review of "Targeting 5-HT Is a Potential Therapeutic Strategy for Neurodegenerative Diseases"

_ijms, 2024, doi:10.3390/ijms252413446_

Round 1

Reviewer 1 Report

Comments and Suggestions for Authors

The review was enjoyable for reviewer who study neurodegenerative diseases and teach pharmacology. This is a comprehensive review with a positive perspective. However, it is pointed out below that it is too exhaustive and may contain descriptions of articles that are somewhat uncertain, only indirectly relevant, or may be mixed with claims by the author that are not well evidenced.

However, as noted above, I want the authors to make the review even easier to read because of its interesting perspectives. I thought the author could have been bolder in focusing on the topic and describing credible information, as describing all relevant topics tends to make it messy and difficult to read.

As a general physiological function, there should be a little more explanation about 5HT1 receptors, such as how they negatively regulate 5HT and other neurotransmitters, such as autoreceptors or heteroreceptors.

The author can decide whether to include this information in the paper, but here, I will provide some information on approved drugs. I think it is more reliable to base the discussion on approved drugs.

l   Lasmiditan was approved by the FDA in 2019 as a treatment for migraine, and it is a 5HT1F agonist. I am unsure if it is worth adding to this review, but just for the author’s reference. I think that it is a drug that specifically suppresses the expression of calcitonin gene-related protein (CGRP), so it is less likely to cause cerebrovascular side effects.

l   They have a mixed effect of various serotonin antagonisms (or partial agonist action), but it has been reported that vortioxetine and lurasidone have 5HT7 antagonism.

AD drugs are better described in more specific terms, such as antibodies against aggregated Aβ, BACE inhibitors that inhibit Aβ production, and tau aggregation inhibitors.

I think the authors need to be more creative with their notation, because brexpiprazole is interpreted as a pure 5HT2A receptor antagonist in the way this review is written. The pharmacological actions are more complex. In addition, the FDA has approved brexpiprazole for the treatment of behavioral disorders in AD.

The talk about 5HT4 receptor agonists and PD is a bit of a digression. The talk about ileus is about the enteric nervous system, so it's not easy to connect it to the central nervous system. If authors do include it, they will probably need an introduction that makes it easier to understand. Personally, I did not think that it was necessary to include it. To confirm, is the talk about serotonin receptors in section 3.2, limited to PD? I feel like there are other talks that seem to be related to other NDs scattered about.

The section describing PD's movement disorders, the effect of 5HT drugs (?) on vomiting is written in a rather vague way, and it feels like it is off topic. It would be better to leave this out, as discussing non-essential points gives the impression of being a jumble.

Quotes 153 and 154 are not about ALS; are they? The author's assertions involved in these quotes seemed to be simply the author's opinion, and it would be better not to include that there is not much evidence directly related to ALS.

Quote 157 likewise seems somewhat over-discussed; perhaps it would be better to provide direct evidence for the psychiatric symptoms of ALS. This topic probably applies to most NDs.

Citation 166 in the HD section is quite old, and case 1.

As a review paper, it would be better to select reliable data rather than too comprehensive. However, if the authors have a basis for trusting the data, then it is fine as it is.

Figure 1 may have a slightly lower resolution.

Figure 2 shows unnecessary commas. In addition, given that brexpiprazole is approved, it would be better to include 5HT2A or 5HT1A partial agonist action in the treatment of AD.

Table 1 would be easier to read if each item were separated by a vertical line. Also, I think it is somewhat awkward that the item names are on two lines.

Author Response

We deeply appreciate your thorough analysis and for raising these concerns. Your insights are invaluable for enhancing the overall quality of our manuscript, and we hold your opinions in high regard. In response, we have made every effort to address your concerns. Please find the detailed responses in the attachment. 

Reviewer 2 Report

Comments and Suggestions for Authors

Xing and co-workers summarized the potential of targeting 5-HT in the therapy of various neurodegenerative disorders. Their detailed review provides a good insight into the possibilities and challenges in this field.

There are only a few remarks/questions.

The figures and the table are relay helpful to follow the article. However, a table/figure concerning the 5-HT receptors and their possible functions, known agonist/antagonist may also help the readers to better follow the review.

The authors mention that “levels of tryptophan metabolites such as quinolinic acid and kynurenic acid have been found to correlate with Aß and tau levels, suggesting a close link between tryptophan and the core pathology of AD [8].” Are there data concerning the level/activity of receptors in the affected brain regions of AD patients?

In PD, aggregation of alpha-synuclein as the hallmark of this disease should be mentioned.

Author Response

We are sincerely grateful to your positive comments. In response, we have carefully revised the manuscript based on your suggestion, and the modifications have been highlighted in the revised manuscript.

Point-to-point replies to the reviewers’ comments:

Reviewer 2:

Comments and Suggestions for Authors

Xing and co-workers summarized the potential of targeting 5-HT in the therapy of various neurodegenerative disorders. Their detailed review provides a good insight into the possibilities and challenges in this field. There are only a few remarks/questions.

Response: We are sincerely grateful to your positive comments. Over the past few days, we have carefully revised the manuscript based on your suggestion, and the modifications have been highlighted in this revised manuscript.

Question 1 (Q1): The figures and the table are relay helpful to follow the article. However, a table/figure concerning the 5-HT receptors and their possible functions, known agonist/antagonist may also help the readers to better follow the review.

Response1: We appreciate your incredibly helpful suggestion. We have reorganized the section on 5-HT receptors and their possible functions, known agonists/antagonists into a new table (Table 1) on page 5. This table has made our content more systematic and intuitive.

Table 1. 5-HT receptor, function, and known agonists/ antagonist

5-HT receptors

possible functions

known agonist/antagonist

5-HT1A

anxiety, mood, cognition, suicidal tendencies, appetite, sleep, and pain perception

Buspirone

5-HT1B/1D

stress responses, mood, and motor control

Naratriptan

Sumatriptan

5-HT1F

pain

Lasmiditan

5-HT2A

cognition, mood, perception, and behavior

Risperidone

Olanzapine

5-HT2C

appetite regulation, weight control, mood and behavior regulation

Lorcaserin

5-HT3

vomiting reflex, cognition, and anxiety

Ondansetron

Granisetron

Palonosetron

5-HT4

neurotransmission, gastrointestinal motility and secretion, and memory

Cisapride

Mosapride

Tegaserod

5-HT6

memory, energy metabolism, and mood

AVN-211

5-HT7

learning and memory, mood regulation, sleep, addictive behaviors, and pain

SB-269970

SB-656104-A

Q2: The authors mention that “levels of tryptophan metabolites such as quinolinic acid and kynurenic acid have been found to correlate with Aß and tau levels, suggesting a close link between tryptophan and the core pathology of AD [8].” Are there data concerning the level/activity of receptors in the affected brain regions of AD patients?

Response2: We sincerely appreciate your concerns. After further review, we found that there is limited evidence directly linking the receptors for quinolinic acid and kynurenic acid to AD patients. Therefore, we believe it would be more accurate to omit references to tryptophan metabolites in this context. In light of this, we have revised the sentence on page 5 from: “levels of tryptophan metabolites such as quinolinic acid and kynurenic acid have been found to correlate with Aß and tau levels, suggesting a close link between tryptophan and the core pathology of AD [8]” to “In AD patients, there is a complex and regionally specific alteration in the expression and function of 5-HT receptors, these changes contribute to both the cognitive and neuropsychiatric symptoms of AD.” We hope this revision addresses your concern effectively.

Q3: In PD, aggregation of alpha-synuclein as the hallmark of this disease should be mentioned.

Response3: Thank you very much for pointing this out. This was indeed an oversight of the previous manuscript to miss such important information. We have added the description as “Additionally, the aggregation of alpha-synuclein is a key pathological feature of PD, contributing to the formation of Lewy bodies[97]” on page 7.

Round 2

Reviewer 1 Report

Comments and Suggestions for Authors

The author has sincerely addressed my concerns.

I think this is a very interesting and comprehensive review.